# Dual Laser Beam Asynchronous Dicing of 4H-SiC Wafer

**DOI:** 10.3390/mi12111331

**Published:** 2021-10-29

**Authors:** Zhe Zhang, Zhidong Wen, Haiyan Shi, Qi Song, Ziye Xu, Man Li, Yu Hou, Zichen Zhang

**Affiliations:** 1Microelectronics Instruments and Equipment R&D Center, Institute of Microelectronics, Chinese Academy of Sciences, Beijing 100029, China; zhangzhe1@ime.ac.cn (Z.Z.); wenzhidong@ime.ac.cn (Z.W.); shihaiyan@ime.ac.cn (H.S.); liman@ime.ac.cn (M.L.); 2School of Microelectronics, University of Chinese Academy of Sciences, No. 19(A) Yuquan Road, Beijing 100049, China; 3International Research Centre for Nano Handling and Manufacturing of China, Changchun University of Science and Technology, Changchun 130022, China; songqi@ime.ac.cn (Q.S.); xuziye@ime.ac.cn (Z.X.)

**Keywords:** silicon carbide, wafer dicing, stealth dicing, laser thermal separation, dry processing, laser processing

## Abstract

SiC wafers, due to their hardness and brittleness, suffer from a low feed rate and a high failure rate during the dicing process. In this study, a novel dual laser beam asynchronous dicing method (DBAD) is proposed to improve the cutting quality of SiC wafers, where a pulsed laser is firstly used to introduce several layers of micro-cracks inside the wafer, along the designed dicing line, then a continuous wave (CW) laser is used to generate thermal stress around cracks, and, finally, the wafer is separated. A finite-element (FE) model was applied to analyze the behavior of CW laser heating and the evolution of the thermal stress field. Through experiments, SiC samples, with a thickness of 200 μm, were cut and analyzed, and the effect of the changing of continuous laser power on the DBAD system was also studied. According to the simulation and experiment results, the effectiveness of the DBAD method is certified. There is no more edge breakage because of the absence of the mechanical breaking process compared with traditional stealth dicing. The novel method can be adapted to the cutting of hard-brittle materials. Specifically for materials thinner than 200 μm, the breaking process in the traditional SiC dicing process can be omitted. It is indicated that the dual laser beam asynchronous dicing method has a great engineering potential for future SiC wafer dicing applications.

## 1. Introduction

SiC power devices have continuously increased their share in the high-power semiconductor market in the last decade and are used in a series of applications such as electric vehicles and urban rail transit. However, due to their hardness and brittleness characteristics, there is one bottleneck in the SiC device manufacturing field, which is the wafer dicing process. Currently, SiC wafers are mainly mechanically diced by diamond-coated blades with low feed rates, in the range of 5–10 mm/s, and a high risk of side chipping at the edges of the diced chips. Furthermore, the diameter of the 4H-SiC wafer has increased from 25 to 100 mm, and the 150 mm transition is upcoming. With the recent smaller and thinner trend in semiconductor manufacturing, mechanical sawing has reached its limit in SiC wafer dicing.

To improve the dicing quality of SiC wafers, many novel dicing technologies have been developed to fulfill the requirements of throughput, edge quality, and costs, such as laser ablation cutting, plasma cutting, high-pressure water cutting, electrical discharge wire cutting, and water-jet guided laser cutting. The thermal separation method is a critical technology that is suitable for the dicing of hard-brittle materials [1]. Typically, the generation and extension of cracks in materials are critical issues in the field of materials science and engineering [2]. However, the thermal separation method developed the theory of utilizing crack generation and extension along a predetermined trajectory. The development of the theory in this field will further enrich the fracture behavior of micro-nano manufacturing. Based on this method, green and efficient cutting technologies for a wide range of materials have been developed [3].

Currently, there are two major processes based on the thermal separation method: non-premade trajectory cutting (NPTC) [4] and premade trajectory cutting (PTC) [5]. The process of NPTC is: firstly, prefabricate a micro notch at one point on the edge of the sheet; then, the thermal stress generated by the heat source scanning drives the force for crack growth along the scanning track until the whole sheet is fractured. The process of PTC is: firstly, a depth of cutting trajectory is performed on the upper or lower surface of the sheet; then, the thermal stress generated by the heat source scanning drives the cutting track to extend to the depth of the plate.

NPTC is mainly used for the rough machining of thicker glass and ceramic plates and other thick materials, with no chips and microcracks in the middle of the cut trajectory. The objectives of the researchers are to increase the cutting speed, reduce the trajectory deviation, and improve the surface cutting quality. A practical method based on the principle of the NPTC to increase the cutting speed is to change the shape of the heat source energy distribution and application of cooling. Yamamoto et al. [6] used the elliptical distribution CO_2_ laser + water-cooled method for the thermal fracture cutting of glass. It was shown that increasing laser heat source power or applying cooling measures could increase the stress intensity factor at the initial crack to reach the threshold value quickly, to increase the cutting speed. Abramov et al. [7] of Corning used laser-induced thermal cracking to cut chemically strengthened glass. Another method to improve the cutting speed is to use a body heating source. The researchers of the LEMI company from Japan used a surface heating source—a tubular infrared lamp with an output power of 1 kW (for body heating) was placed 200 mm above the material and superimposed on the LD laser scan. However, the cutting speed without the IR lamp was only 23 mm/s [8].

The major problem in NTPC is trajectory deviation. Salman et al. [9] used a diode laser to cut 5-mm-thick soda-lime glass at a speed of 33 mm/s. It was found that there were severe trajectory shifts at the entrance and exit of the material. They simulated the stress field of the workpiece during the cutting process. It was found that the reason for the shifted trajectory was that the tensile stress at the entrance and exit of the cut was huge. Salman et al. [10] also studied the stress distribution at the entrance and exit of the cut using simulation. The artificial neural network model and finite element model were applied, respectively, for different thicknesses and laser scanning speeds. The results showed that the prediction results of the artificial neural network model were better than those of the finite element model.

The surface quality obtained by the thermal cracking method effectively improves bending strength. Kondratenko et al. [11] investigated laser-induced thermal cracking of cutting glass with thicknesses of 4–19 mm. They compared the strength of 6-mm-thick glass cut by mechanical, grinding, and laser. Finally, the quality of glass cut by the laser-induced thermal cracking method was better, and the edge strength was 5.5 times higher than that of conventional mechanical cutting.

Premade trajectory cutting is mainly used for the processing of liquid crystal display (LCD), plasma display (PDP), and flat panel display (FPD). PTC is characterized by faster cutting speeds and higher accuracy of the cutting trajectory. Many researchers have expanded the applicability of the PTC method for material dicing. Kang et al. [12] performed high-speed cutting of laminated glass. For PDP cutting, the authors used a two-step cutting method of scribing and thermal cracking, which is more efficient than the one-step cutting method. A series of processing devices were developed based on this principle. Huang et al. [13] created an implicit crack inside the glass using a 10 W 355 nm Nd:YAG laser. Huang et al. [14] also introduced ultrasonic vibration into the Nd:YAG UV laser and continuous CO_2_ laser in the cutting system of LCD glass substrates. The results showed that the introduction of ultrasonic vibration could improve the cutting speed greatly (three times the original speed).

Cross-sectional quality is significantly improved by using PTC. KIM et al. [15] used a femtosecond laser for etching and a CO_2_ thermal laser for cracking to separate LCD glass. The experimental results showed that at low numbers of femtosecond laser pulses, the glass damage was small and the groove depth was not deep enough. When the number of femtosecond laser pulses was increased to six, the following CO_2_ laser was more effective in separating the glass by thermal stress. Wang et al. [16] conducted a study on laser thermal cleavage-cutting crystal glass substrates and proposed a new grooving and thermal cracking cutting method. Firstly, micro-cracks were created on the surface of liquid crystal glass using the instantaneous high energy of a YAG laser. Then, the glass was heated by CO_2_ laser and cooled by Ar gas.

Stealth dicing [17,18,19] is another state-of-the-art dicing method where a pulsed laser, at a wavelength capable of penetrating the material, is focused inside the substrate. Focused laser spots cause an extremely high power density, both temporally and spatially, at localized points. By moving the laser along the desired path at different depths, several passes of the laser ablation points are formed. When external tensile stress is subsequently applied, the dies are separated. The process is fast, clean, and has zero kerfs. However, stealth dicing is typically combined with a mechanical breaking process. When it comes to hard-brittle material such as SiC [20], this mechanical breaking process can cause serious edge breakage and even cause wrong crack propagation and, finally, reduce the dicing yield.

In this work, we investigate a dual laser beam asynchronous dicing method by combining stealth dicing and premade trajectory cutting. The laser-based cutting method proposed in this work is clean, fast, and efficient and does not involve any chemical agent or liquid. The dependence of the laser process parameters on the cutting quality was theoretical and experimentally investigated. The quality of the cut edge was thoroughly analyzed by optical microscopy.

## 2. Materials and Methods

Figure 1 shows the whole set-up used in this study, which includes a 5 W-532 nm femtosecond pulsed laser manufactured by NKT working under the condition of 750 fs pulse duration, and a 10 W-1040 nm CW laser manufactured by IPG with customized wavelength. The pulsed laser is focused on the interior of the SiC wafer by an objective microscope lens to create bottom-up stealth dicing layers, while the continuous laser is sent to a Galvano scanner system and used as a heat source to create thermal stress.

### 2.1. Dicing Method

The operation of the DBAD method in our experiments is as follows: firstly, stealth dicing is operated on SiC. The pulse duration of the laser used in this process is 750 fs, and the scanning speed is 3000 μm/s. Through the fine control of focus depth and single pulse energy, the surface has no visible cracks after the process. Then, a CW laser sweeps through the trace of SD, and the internal cracks from the stealth dicing are extended vertically due to the thermal stress. The parameters of the two lasers are listed in Table 1.

### 2.2. Samples

Briefly, 4-inch diameter silicon carbide (4H-SiC) wafers of 200 μm thickness were selected in this study; the physical properties of the material are shown in Table 2. The key principle of this dicing method is to minimize cracks and chippings and realize the cutting track as straight as possible. As a result of the wafers’ brittle and hard characteristics, different laser parameters lead to very different results.

### 2.3. Numerical Molding

To better understand the heat accumulation process and stress concentration process caused by the continuous laser, a two-dimensional finite element model (FEM) was established. The thermal stress produced by the moving CW laser around the interior hole produced by the previous pulsed laser is shown schematically in Figure 2. The height of the voids produced by the pulse laser set in this model is 20 μm, while the width is 5 μm. The size is approximately the same as the size of the hole created in the experiment.

High-intensity lasers, incident upon a material that is partially transparent, will deposit power into the material itself. The absorption of the incident light can be described by the Beer–Lambert law as it can be written in differential form for the light intensity I as:∂I/∂z = α(T)I,(1)
where z is the coordinate along the beam direction, and α(T) is the temperature-dependent absorption coefficient of the material. As the heating and subsequent cooling process can vary in space and time, the evolution of temperature distribution is predicted by solving the time-dependent partial differential equation:ρC_p_∂T/∂t − ∇⋅(k∇T) = Q = α(T)I,(2)
where Q is the heat source, which is equal to the absorbed light; ρ and C_p_ are the density and constant pressure heat capacity of the material, respectively. The formula of thermal stress is given by:F = Y(ε∆T)/L_0_,(3)
where Y is Young’s modulus of the given material, ε is the coefficient of linear thermal expansion of the given material, and L_0_ is the original length of the material before the expansion. These three equations are coupled with each other and are resolved using COMSOL Multiphysics.

The component structure with mesh is shown in Figure 3. A cuboid model is established with three microvoids inside, which are shaped like an ellipsoid to simulate the micro-cracks ablated during the stealth dicing process. For balancing the demand for simulating precision and computational efficiency, the model is simplified to a mirror-symmetrical model in the x–z plane, and infinity element layers are used for thermal diffusion simulation of large wafers. The energy deposition is assumed to be a moving Gaussian profile and is modeled by a boundary heat flux in the x–y plane (z = 0); energy depositions generate heat in the material, which can cause local stress concentrations resulting from thermal expansion.

## 3. Results

### 3.1. Numerical Molding Results

Figure 4 shows the dynamic temperature change of the 4H-SiC material during CW laser scanning when P = 10 W, v = 1000 mm/s. The maximum temperature inside the sample is only 312 K, which is far below the melting point of 4H-SiC. It can be concluded that the CW laser scanning process does not induce thermal damage to the substrate.

Figure 5 shows the thermal stress at the endpoint of the long axis of the void-changing process during laser scanning. The maximum thermal stress, 48 MPa, presents at t = 400 μs, which indicates that 10 W of CW laser power is capable of generating enough thermal stress for the separation process.

Figure 6 shows the thermal stress distribution change along the z-direction through the center of three voids, as we can see a large stress gradient at both the endpoints of the long axis of the ellipsoid; this is the main cause for the cracks to spread along the z-direction.

### 3.2. Experimental Results

In this study, a series of experiments were conducted to cut SiC using a dual laser beam asynchronous dicing method based on the finite element simulation. Based on the principle of DBAD, single-pass stealth dicing is processed on the material, and then the thermal stress is generated using the 8 W-1040 nm continuous laser to extend the crack. Finally, the material is cut completely through the simple wafer expanding process. If the stealth cutting operation and thermal cracking are performed simultaneously, it will lead to misalignment and defects inside hard-brittle materials such as SiC, resulting in large errors in the positioning accuracy of the subsequent process.

Figure 7 shows the experimental results of cutting the SiC wafer with a thickness of 200 μm using DBAD compared with the simulation results. The surface of the material after stealth dicing is illustrated in Figure 7a. There were processing traces with no remarkable cracks. Then, a clear crack was performed through scanning using the continuous laser due to the thermal separation, as shown in Figure 7b. Figure 7c shows the cutting profile of SiC with a thickness of 200 μm. There was a line of three craters because of the process of SD. The height of the craters was about 25 μm, and the width was almost 9 μm. Additionally, it can be seen that the cracks from the thermal press could almost separate the material. Finally, in Figure 7d, the wafer of SiC is completely separated after the simple wafer-expanding process.

The experimental results are in agreement with the above FEM simulation results, proving the correctness of the FEM simulation model and the effectiveness of the DBAD method. The quality of processing is perfectly expected. Compared with traditional stealth dicing, there was no serious edge breakage and no wrong crack propagation occurred from the mechanical breaking process. During the stealth dicing operation, there was no remarkable crack trace on the surface, so that the processing window was expanded with an improved production rate. In addition, the following wafer-expanding process maybe not be necessary, especially for the processing of thin hard-brittle materials. It is also suitable for the processing of hard-brittle materials with a thickness less than 200 μm.

## 4. Discussion

In our experiments, the effect of the changing continuous laser power on the DBAD system was also studied. The thermal press is generated using a 1040 nm continuous laser. However, the crack trajectory may be bent with the power of the laser rising to 10 W compared with the crack using an 8 W continuous laser, as seen in Figure 8. The reason for this phenomenon is still unknown. The effect of continuous laser parameters on the DBAD process will be investigated deeply in subsequent studies. It is beneficial to find better laser parameters during the processing.

Miniaturization and performance enhancement drives the development of wafer separation technologies. Especially for hard-brittle materials, chippings occur with a poor cutting quality using mechanical dicing, while traditional stealth dicing with a mechanical breaking process leads to edge breakage. It is shown that the validity of the DBAD method is certified because of the above finite element simulation and experiments. The processing flow for cutting hard-brittle materials is simplified by improving processing quality compared with traditional processing. An effective method is provided, using dual laser beam asynchronous dicing to cut hard-brittle materials such as SiC.

## 5. Conclusions

In this paper, a dual laser beam asynchronous separation method for SiC wafer dicing has been put forward. In this method, a series of micro-cracks is firstly formed inside the wafer through a stealth dicing process by controlling the focal depth; the SD process will not induce visible cracks on the surface. Then, a CW laser is loaded on the dicing street, and the thermal stress leads to the wafer separation process.

The absorption of the CW laser and the resulting thermal stress was calculated using a finite element model. The simulation indicated that the tensile stress produced by the CW laser heating in the upper and lower ends of the voids is the main mechanism of vertical crack propagation. To get better separation quality, the moving speed and the power of CW laser should be properly adjusted. Insufficient laser power will weaken thermal accumulation, and the SiC wafer will not reach its fractural strength. In contrast, excess laser power can cause unnecessary heat accumulation and potentially harm the circuit. Based on the simulation analysis, an experimental machine was built, and several 4H-SiC wafers with a thickness of 200 μm were cut; a neat cutting side wall without chipping was obtained. Through the experimental study, it can be determined that the proper depth of the last SD layer can be 50 μm below the surface, and the optimal moving speed of the CW laser is 1000 mm/s; acting with 8 W of laser power, the fracture will propagate upward stably, and the SiC wafer can be separated along the expected SD path.

This novel DBAD method provides an effective solution for wafer cutting, specifically for hard-brittle materials with a thickness less than 200 μm, compared with other traditional processing methods.

## Figures and Tables

**Figure 1 micromachines-12-01331-f001:**
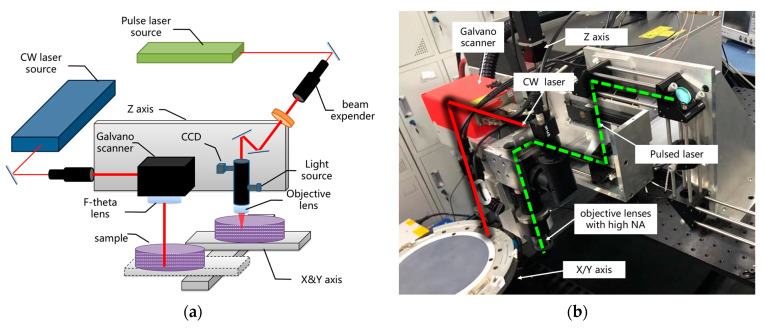
Configuration of a dual laser beam asynchronous dicing system: (**a**) schematic diagram of experimental set-up; (**b**) experimental platform with dual laser path.

**Figure 2 micromachines-12-01331-f002:**
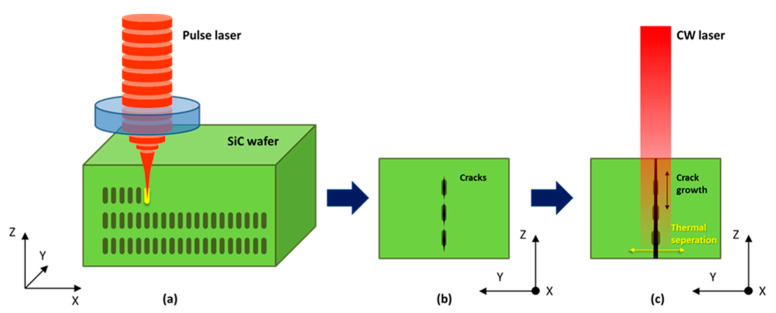
The schematic of the FEM model. (**a**) The stealth dicing process with the pulse laser, (**b**) the internal void produced from (**a**,**c**). The crack growth process caused by the CW laser.

**Figure 3 micromachines-12-01331-f003:**
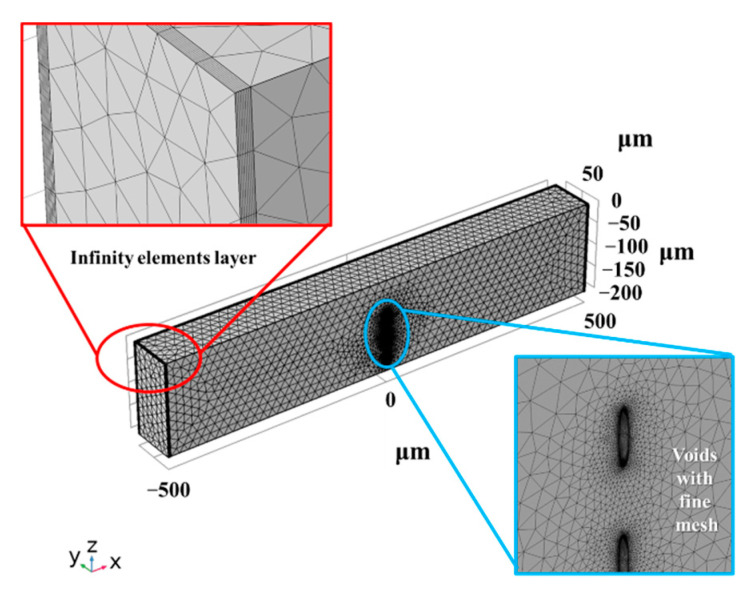
The diagram of the component structure with mesh.

**Figure 4 micromachines-12-01331-f004:**
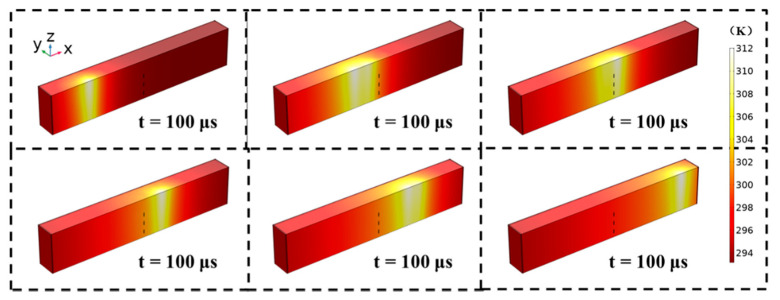
Moving laser heating material (10 W; 1000 mm/s).

**Figure 5 micromachines-12-01331-f005:**
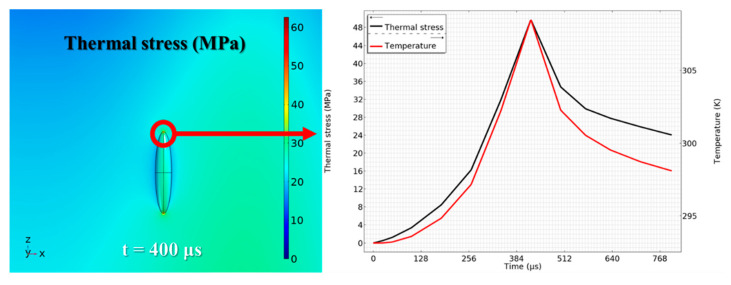
Thermal stress at the endpoint of the long axis of voids.

**Figure 6 micromachines-12-01331-f006:**
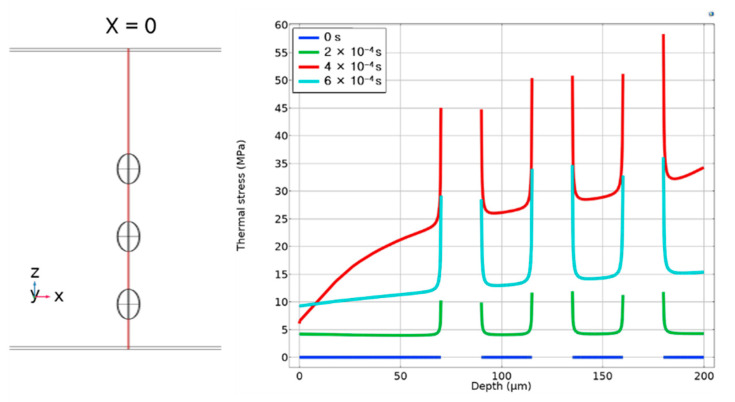
Thermal stress distribution along the z-direction.

**Figure 7 micromachines-12-01331-f007:**
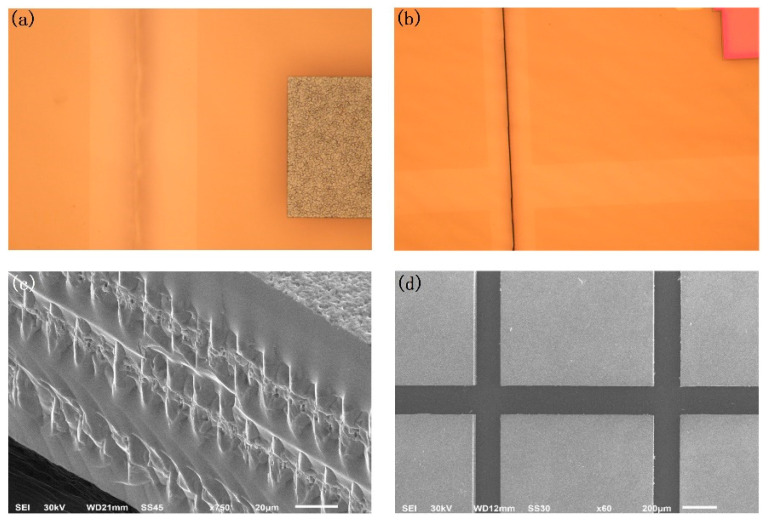
The results of cutting 200 μm SiC. (**a**) The surface after the stealth dicing; (**b**) the thermal crack from the continuous laser; (**c**) the cutting profile; and (**d**) the final separation after the expanding process.

**Figure 8 micromachines-12-01331-f008:**
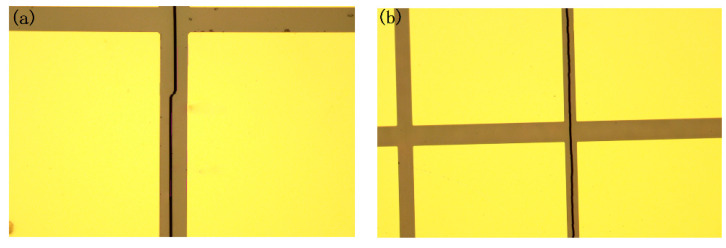
The crack from cutting the 200 μm SiC using (**a**) a 10 W/1040 nm continuous laser and (**b**) an 8 W/1040 nm continuous laser.

**Table 1 micromachines-12-01331-t001:** Main parameters of the lasers.

Laser Parameters	Pulsed Laser	CW Laser
Wavelength	532 nm	1040 nm
Max power	5 W	10 W
Repetition rate	20~200 kHz	N/A
Focal length	4 mm	140 mm
Beam diameter	8 μm	20 μm
Quality factor	M2 < 1.1	M2 < 1.2
Beam mode	TEM_00_ Gaussian	

**Table 2 micromachines-12-01331-t002:** Physical properties of 4H-SiC.

Material Properties	Value
Density	3210 kg/m^3^
Thermal conductivity	490 W/(m⋅K)
Constant pressure heat capacity	690 J/(kg⋅K)
Coefficient of thermal expansion	4.3 × 10^−6^ 1/K
Poisson’s ratio	0.185
Young’s modulus	7 × 10^11^ Pa
Absorption coefficient	30 cm^−1^ [21]

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
