# Peer review of "Dual Laser Beam Asynchronous Dicing of 4H-SiC Wafer"

_micromachines, 2021, doi:10.3390/mi12111331_

Round 1
Reviewer 1 Report
A dual laser beam asynchronous separation method for SiC wafer ducting has been work out.
In my opinion this method can be efficiently applied in practice.
The methodology of research is appropriate for case taken into account.
At the beginning The Authors took into accont (in wide range) informations from published papers then go on with working out the physical and mathematical modeles of the investigated process. The results of calculations have been veryfied in experimental way,
The experimental research have been planned and did in right way, The methodology of experiments was right. Technical level of test stand was acceptable. Methods and methodology of measurements were wright.
Analysis of phenomena occuring in machining area and analysis of experimental test results were done in professional way. They are the base for right final conclusions.

Author Response
Dear reviewer,
Thank you for reviewing our manuscript and for the sincere comments. We have revised our manuscript minorly. The revised manuscript has been uploaded to the attachment. Thanks again for supporting our manuscript.

Reviewer 2 Report
The article titled “Dual Laser Beam Asynchronous Dicing of 4H-SiC Wafer” reports the advantage of the dual laser beam synchronous dicing method for the quality of the cutting edge by eliminating the mechanical breaking process in Stealth Dicing and replacing it with a thermal breaking process. The introduction is carefully written for easy understanding by non-specialists, and the experimental and calculation results are clear. On the other hand, improving the description of the details of the calculation and experimental conditions is desirable.
Though the absorption coefficient of the substrate is explained in Equation 1, there is no explanation of the value used in the calculation. The values or references should be described.
The information about the size of the holes placed in the calculation is not described. It is possible to read the major axis from Fig. 6. However, it would be helpful to describe it explicitly in the main text. Is it confirmed that the size of the hole used in the calculation is approximately the same as the size of the hole created in the experiment?
The conditions for creating a void with the laser pulses are not described. It would be desirable to describe the time required to form a single void or the scan speed to create voids along the trajectory path.
Sentence of Fig 2 caption is right? Please check figure captions. It is recommended that figure captions be described in more detail.
“Baes on” in line 205 is probably a typo for “Based on”.
Author Response
Dear reviewer,
Thank you for reviewing our manuscript and for the constructive comments, which greatly helped us to improve the manuscript.
(Manuscript ID: micromachines-1424370
Type of manuscript: Article
Title: Dual Laser Beam Asynchronous Dicing of 4H-SiC Wafer)
We have studied comments carefully and have made correction which we hope meet with approval. Revised portion are marked in red in the paper. The main corrections in the paper and the responds to your comments are as follows:
1, The absorption coefficient of the substrate
Response: The absorption coefficient of the 4H-SiC used in the calculation is 30cm-1. This value is refer to the article (Biedermann, E. The optical absorption bands and their anisotropy in the various modifications of sic. Solid State Communications 1965, 3, 343-346.).
2, The size of the holes
Response: The height of the voids produced by the pulse laser set in this model is 20μm while the width is 5μm. The size is approximately the same as the size of the hole created in the experiment. In the experiment, the height of the creaters is about 25μm and the width is 9μm almostly.
3, The conditions for creating a void with the laser pulses
Response: The pulse duration of the laser used in the stealth dicing process is 750fs and the scanning speed to create voids along the trajectory path is 3000μm/s. Through the finely control of the focus depth and single pulse energy, the surface has no visible cracks after the process.
4, Sentence of Fig 2 caption
Response: Thank you so much for your careful check. The caption for Fig 2 have been corrected as follows:
The schematic of the FEM model. (a) The stealth dicing process with the pulse laser, (b) The internal void produced from (a) and (c) The crack growth process caused by the CW laser.
5, The “Baes on” has been revised as “Based on”.
Thanks again for your time spend making your constructive remarks and suggestions, which has significantly raised the quality of the manuscript.
The revised manuscript has been uploaded to the attachment.
